# Quantitative Analysis of Major Metals in Agricultural Biochar Using Laser-Induced Breakdown Spectroscopy with an Adaboost Artificial Neural Network Algorithm

**DOI:** 10.3390/molecules24203753

**Published:** 2019-10-18

**Authors:** Hongwei Duan, Lujia Han, Guangqun Huang

**Affiliations:** Laboratory of Biomass and Bioprocessing Engineering, College of Engineering, China Agricultural University, Beijing 100083, China; dhwsg123@cau.edu.cn (H.D.); hanlj@cau.edu.cn (L.H.)

**Keywords:** agricultural biochar, heavy and nutritional metals, LIBS, BP-Adaboost

## Abstract

To promote the green development of agriculture by returning biochar to farmland, it is of great significance to simultaneously detect heavy and nutritional metals in agricultural biochar. This work aimed first to apply laser-induced breakdown spectroscopy (LIBS) for the determination of heavy (Pb, Cr) and nutritional (K, Na, Ca, Mg, Cu, and Zn) metals in agricultural biochar. Each batch of collected biochar was prepared to a standardized sample using the separating and milling method. Two types of univariate analysis model were developed using peak intensity and integration area of the sensitive emission lines, but the performance did not satisfy the requirements of practical application because of the poor correlations between the measured values and predicted values, as well as large relative standard deviation of the prediction (RSDP) values. An ensemble learning algorithm, adaboost backpropagation artificial neural network (BP-Adaboost), was then used to develop the multivariate analysis models, which had a more robust performance than traditional univariate analysis, partial least squares regression (PLSR), and backpropagation artificial neural network (BP-ANN). The optimized RSDP values for K, Ca, Mg, and Cu were less than 10%, while the RSDP values for Pb, Cr, Zn, and Na were in the range of 10–20%. Moreover, the pairwise *t*-test of its prediction set showed that there was no significant difference between the measurements of LIBS and ICP-MS. The promising results indicate that rapid and simultaneous detection of major heavy and nutritional metals in agricultural biochar can be achieved using LIBS and reasonable chemometric algorithms.

## 1. Introduction

Agricultural biochar is the main product of the high-temperature pyrolysis of agricultural waste materials under oxygen-limited conditions [1]. Returning biochar to farmland is one of the most effective ways to promote the green development of agriculture [2]. Metal cations on the biochar surface such as K^+^, Ca^2+^, Na^+^, and Mg^2+^ can not only provide abundant potassium fertilizer for farmland soil, but also increase soil pH. Moreover, the porous surface can adsorb heavy metal ions such as Pb^2+^, Cr^2+^, and Cr^3+^ in contaminated soil [3,4]. Therefore, a quantitative analysis of the major metal elements in agricultural biochar is necessary to monitor farmland environment and ensure food crop safety.

Laser-induced breakdown spectroscopy (LIBS) has been proven to be a rapid and minimally destructive analytical method for almost total elemental analysis of various materials with little or no sample pretreatment [5] versus classic spectrometric techniques, such as atomic absorption spectrometry (AAS) [6], inductively coupled plasma atomic emission spectrometry (ICP-AES) [7], and inductively coupled plasma mass spectrometry (ICP-MS) [8]. LIBS has been widely used in industrial field analysis [9,10], deep sea exploration [11], environmental monitoring [12], explosives analysis [13,14], biomedicine [15], and agriculture and food [16,17,18]. It thus has great potential to cover the shortage of traditional laboratorial analytical methods in the analysis of agricultural biochar.

Simultaneous quantitative analysis of multi-indicators is the basic goal of agricultural biochar functional assessment. However, the special properties of biochar, such as particle size, heterogeneity, porous surfaces, and complex components introduce more complicated matrix effects to the implementation of LIBS, posing a great challenge for the measurement stability and detection accuracy [19,20]. In recent years, an enormous amount of research effort has gone into upgrading LIBS equipment to improve analytical performance, resulting in variants such as spatial confinement LIBS [21], femtosecond LIBS [22], double-pulse LIBS [23], laser-induced fluorescence-assisted LIBS [24], microwave-assisted LIBS [25], and spark discharge LIBS [26]. However, these methods inevitably increase the instrument complexity and economic burden. Alternatively, chemometrics coupled with LIBS has become another hot topic in the academic community, which can be implemented with the goal of extracting feature information by eliminating noise information from the acquired spectra. In particular, multivariate analysis can not only improve the detection accuracy, but also reduce the detection limit. Guo et al. [20] applied LIBS coupled with a nonlinear multivariate regression support vector machine method to accurately quantify twelve elements, excluding Cu, in soil. Yao et al. [27] pointed out that the detection results of linear multivariate analytical method of partial least squares regression (PLSR) had better consistency with AAS for Pb and Cr in pork than the traditional univariate calibration curve method. In our previous work [28], an improved wavelet dual threshold function was proposed and used with the PLSR method to accurately detect the Cu and Zn content in Chinese animal manure compost.

The potential of LIBS for the quantitative analysis of major harmful and nutritional metals in agricultural biochar was explored in this study. To overcome the complicated matrix effect in agricultural biochar, an ensemble learning algorithm, adaboost backpropagation artificial neural network (BP-Adaboost), was implemented to develop the nonlinear multivariate regression model, and its performance was compared with traditional backpropagation artificial neural network (BP-ANN) and linear calibration methods, including univariate analysis and PLSR. The determinant coefficients, root mean square errors, and relative standard deviations of the three methods were compared, and the best scheme proposed to assess biochar quality.

## 2. Results and Discussion

### 2.1. Statistical Analysis of Elemental Contents in Standard Samples

Table 1 summarizes the statistical results of the major metal elements (Pb, Cr, Cu, Zn, K, Na, Ca, and Mg) of the prepared standard biochar samples, including the range, mean, and standard deviation (SD). The samples S1 to S9 were selected as the calibration set, while the samples S10 to S18 were considered the prediction set. Some samples in the prediction set were excluded because their elemental content exceeded the range of the calibration set. The averaged content of Pb was almost twice of the content of Cr. Moreover, the averaged content of K, which can improve plant photosynthesis, was the highest among the nutritional elements.

### 2.2. Spectral Analysis

The averaged spectra of rice husk, rice straw, and corn stalk biochar in the wavelength range of 187.78–982.29 nm are presented in Figure 1. A similar tendency was identified from the three LIBS spectra, which signified that these biochar may have the same elemental species and similar chemical components. In the three kinds of biochar, the rice husk biochar had the largest peak intensities of K (located at 766.49 nm and 769.90 nm) and Na (located at 588.99 nm and 589.59 nm), but the lowest concentrations for these two elements. This may be explained by the existence of a relatively weak matrix effect at these locations for rice husk biochar, resulting in the relatively large intensities.

Based on the atomic spectral database (ASD) of the National Institute of Standard and Technology (NIST), the emission lines of the major metal atoms and ions were confirmed and are presented in Table 2. The peak intensities of K, Na, Ca, and Mg were larger than those of Pb, Cr, Cu, and Zn because of their lower ionization energies. Ca had saturated peaks at 393.37 nm, 396.85 nm, and 422.67 nm, which were not used for the development of calibration models.

Principal component analysis (PCA) can be used for sample classification and outlier elimination based on the score matrix. The PCA results for the eighteen spectra of agricultural biochar are presented in Figure 2a. The wide distribution of the score matrix indicated the representativeness of collected samples [29]. According to the PCA plot, the first three PCs explained a total of 90% of the variation for the full spectral data, including 57.01%, 21.22%, and 12.13% for PC1, PC2, and PC3, respectively. Furthermore, the three kinds of biochar had their own distinct characteristics on PC2, and variables (Ca II 393.37, Ca II 396.85 nm, Ca I 422.67 nm, Ca I 612.22 nm, Ca I 616.12 nm, Al I 394.40 nm, Al I 396.15 nm, Mg II 279.55 nm, Mg I 383.20 nm, Ti I 443.41 nm, Ti I 445.42 nm, K I 766.49 nm, K I 769.90 nm, Na I 588.99 nm, Na I 589.59 nm, CN 388.25 nm) in PC2 had relatively large loadings, marked in Figure 2b. It was evident that these elements were highly consistent with those marked in the averaged spectra of the biochar samples.

### 2.3. Quantitative Analysis

#### 2.3.1. Univariate Analysis

In this work, peak intensities and integration areas of six sensitive emission lines (Pb I 406.32 nm, Cr II 427.11 nm, Cu II 324.70 nm, Zn II 206.20 nm, Ca II 317.91 nm, and Mg I 517.27 nm) excluding self-absorption lines and overlapping peaks, were applied to develop the univariate models. The model results for Pb, Cr, Cu, Zn, Ca, and Mg in biochar are presented in Table 3.

The performances of the developed univariate models presented a large variation between different elements. The best relative standard deviation of the prediction (RSDP) values for Cu and Ca were less than 20%, while the best RSDP values for Pb, Cr, Zn, and Mg were all in the range of 20–40%. This indicated the variated matrix effect on different elemental models of agricultural biochar. For Pb, Cr, Cu, Zn, Ca, and Mg, the univariate models developed by integration areas had lower RSDP values of 24.91%, 23.42%, 9.85%, 27.92%, 16.42%, and 20.50% than those of the univariate models developed by peak intensities. The results showed that integration area is more appropriate to build the univariate model, but the performance still needed to be improved by multivariate methods.

#### 2.3.2. Multivariate Analysis

The peak broadening wavebands as shown in Table 2 were selected as the spectral matrix of multivariate analysis. Multiple preprocessing methods, such as baseline correction (BC), normalization (Norm), and autoscaling (AS), were used for spectral de-noising [28,30]. The preprocessed dataset were then implemented to develop the linear PLSR models and nonlinear models of BP-ANN and BP-Adaboost, and the model results for Pb, Cr, Cu, Zn, K, Na, Ca, and Mg in agricultural biochar are presented in Table 4. To improve the computational efficiency and quantitative accuracy of the BP-ANN and BP-Adaboost algorithms, PCA was used to extract feature variables from spectral data in advance. The parameters of BP-Adaboost were set as follows: the number of weak classifiers was set to 40. The neural network was created and trained using the functions of “newff” and “trainlm”. The transferring functions of “logsig” and “purelin” were used for the hidden and output layers. The prediction error threshold was set to 0.001.

For the heavy metals of Pb and Cr, the linear models of PLSR were initially developed using two latent variables (LVs), yielding RSDP values of 18.86% and 17.41%, while four and three principal components (PCs) were used to develop the nonlinear models of BP-ANN and BP-Adaboost, resulting in RSDPs of 15.46% and 11.63%, 13.33%, and 10.18%, respectively. This indicated that the performance of PLSR was inferior to that of BP-ANN, but the BP-Adaboost model was the most robust and was suitable for use in practical detection applications because of its lower RSDP values and large *p* values (>0.05). These results were expected since the multilayer neural network could project the LIBS spectra to high-dimensional space, and more spectral features could be easily identified from matrix noise. In contrast, a great deal of the noise that existed in the original spectra was easily interpreted as valuable signals during the PLSR modeling process, triggering the over-fitting phenomenon and poor prediction ability. Because of the similar components and matrix effect between the biochar and soil samples, we compared our results with the published findings in soil samples, as shown in Table 5. In the case of Pb, the BP-Adaboost model showed a better performance than that of Wang et al. [31], and was comparable to that of Yu et al. [32]. However, the soil samples used by Yu et al. [32] were soaked in lead nitrate solution, and the Pb content was the highest in all three types of matrix. This not only indicated a minimized matrix effect but also implied higher emission intensities and reduced effects from factors such as baseline deviations and instrument noise. As a result of the disadvantages of the trace content and the weak emission intensities of Pb in soil and biochar samples, the analytical lines may be virtually submerged by noise and may even be unobservable. In the case of Cr, the performance in this work was superior to that of Duan et al. [33] and Fu et al. [34], and was comparable to that of Wang et al. [31]. However, the emission lines of multiple elements were all employed to develop the calibration models for predicting the Cr content, but the sensitive variables with large weight coefficients varied in these three reports. This means that the prediction ability of these models may have been poor since their sensitive variables may not be suitable for soils in different habitats. The performances of the best models in terms of measured vs. predicted values of Pb and Cr in agricultural biochar are plotted in Figure 3a,b.

For the nutritional elements except Na, Mg, and Cu, the RSDP values of BP-ANN were lower than that of PLSR, which decreased by 4.73%, 1.57%, and 0.59% for K, Ca, and Zn, respectively. However, for all the nutritional elements, the BP-Adaboost model had the lowest RSDP values in the three multivariate models. Moreover, the *P* values of the prediction set were all higher than the threshold value of 0.05. This result demonstrates that the BP-Adaboost model had a more robust performance for all nutritional metals, and thus has great potential for practical applications. However, the determination coefficient for Ca was negative as a result of the small distribution range in the prediction set. Meanwhile, the RSDP values of BP-Adaboost models were all less than 15%, except Zn. The reason may have been that the concentration of Zn was “blank” in the content range of 92.3762 mg·kg^−1^–313.2861 mg·kg^−1^, weakening the prediction performance of the entire detection range. As shown in Table 5, the prediction abilities for K, Ca, Mg, and Cu in this work were all superior to those in the related literature [31,33,34,35,36,37], and the performance for Zn was comparable to that of Kim et al. [38]. The reason may have been that the BP-Adaboost algorithm has a better adaptive ability than traditional neural networks and PLSR, avoiding the local optimum and over-fitting phenomenon. The performances of the best models in terms of measured vs. predicted values of Cu, Zn, K, Na, Ca, and Mg in agricultural biochar are plotted in Figure 3c–h.

## 3. Materials and Methods

### 3.1. Experimental Setup

The benchtop LIBS system used in this experiment is shown in Figure 4. Laser pulses at 1064 nm with the maximum frequency and energy of 2 HZ and 100 mJ and a 10 ns pulse width were generated by a solid Q-switched Nd:YAG pulse laser (Avantes, Apeldoorn, Netherlands). The focused laser irradiated and ablated the biochar sample at normal incidence, resulting in a crater and inducing plasma emissions. A collimating lens collected the plasma signal into the fiber transmission channel at a 45°angle relative to the horizontal plane. The optical fiber outlet was connected with a seven-channel spectrometer charge coupled device (CCD) array, with a wavelength range of 187.78 –982.29 nm and a spectral resolution of λ/Δλ = 12,297. The high-definition gray camera was applied to adjust the settings of the electric X–Y–Z stage, since its focal length was consistent with that of the laser, which was also used to monitor the ablation spots in real time.

To solve the impact of energy fluctuation on the intensity of the emission line, the laser energy was set to 30 mJ, the number of single-point laser repeat ablations was three, and the spot size was set at 200 μm. To avoid the bremsstrahlung, the delay time relative to the laser pulse was 0.7 μs to obtain the best spectral intensity and signal to background ratio. Spectra of multiple spots on the sample surface were acquired, and the averaged spectrum was considered the final spectrum of each sample.

### 3.2. Standard Sample Preparation

Eighteen batches of agricultural biochar products (~5 kg for each batch) were collected from Nanjing Zhironglian Technology Co., Ltd. (Nanjing, China), including six rice husk, six rice straw, and six corn stalk samples. Because of the complex thermochemical reactions that occur during the production of biochar, particle-shaped biochar appeared in each batch of samples. Furthermore, the porous surface of particulate biochar more easily absorbs heavy metals, such as Pb and Cr, which leads to a large difference in the content of metal elements with particle size distribution.

To prepare standard biochar samples, the separating and milling method [39] was proposed and applied to divide each batch of biochar sample into eight particle size ranges, and their averaged spectra and elemental contents were determined as the standard sample references. The specific steps for preparing the standard biochar sample were as follows:

(1) Each batch of biochar sample was gradually sieved by vibrating screen (JS14S type, Nanjing, China) at the mesh size (6 mm, 4 mm, 1.43 mm, 0.9 mm, 0.45 mm, 0.3 mm, and 0.18 mm), and each screening time was set to 30 min. The sieved samples were crushed using a pulverizer (WKF-130 type, Weifang, China) with a 75 μm sieve, and the resulting samples were placed in valve bags for use.

(2) ICP-MS (PE NexION 300, Waltham, USA) was then used to measure the Pb, Cr, Cu, Zn, K, Na, Ca, and Mg contents of the digestion solution based on the fitted standard curves, obtaining the major metal content of each batch of biochar sample in the eight particle size ranges. Meanwhile, the prepared samples were taped and compressed at a pressure of 20 T with a tablet press (DY30 type, Tianjin, China). Twelve points for each pellet were ablated by LIBS, and a total of 96 points were acquired for each batch of biochar sample, as shown in Figure 5.

(3) To ensure the spectral representativeness, the averaged spectrum of 96 points was regarded as the standard spectrum for each batch of biochar sample. Moreover, the averaged elemental content of the samples in eight particle size ranges was taken as the representative element content, as shown in Table 6.

### 3.3. Quantitative Methods

Univariate analysis is a traditional calibration curve method used to establish the relation between single emission line and elemental content by curve fitting, and is thus easily affected by the matrix effect [18]. Furthermore, the micro-mechanism of the interaction between laser and materials has not been explained by a systematic theory. Therefore, univariate linear regression may not meet the precision requirements of practical applications.

Multivariate analysis can make the best use of the feature variables from the full wavelengths to build a robust calibration model. In particular, the partial least squares regression (PLSR) method is a perfect combination of multivariate linear regression, canonical correlation analysis, and principal component regression. The best linear calibration model can be rapidly established by synchronously decomposing the spectral matrix and concentration matrix [40]. Backpropagation artificial neural network (BP-ANN) is one of the most widely used neural network models using the error back propagation algorithm, while the adaboost backpropagation artificial neural network (BP-Adaboost) is an ensemble learning algorithm of multiple weak learning classifiers [41]. This overcomes the disadvantages of local optimum and over-fitting of traditional backpropagation artificial neural network. It thus has a better nonlinear fitting performance.

The performance of univariate and multivariate calibration models was evaluated by the determination coefficients of calibration (*R*^2^_cal_) and prediction (*R*^2^_p_) sets, the root mean square errors of the calibration (RMSEC) and prediction (RMSEP) sets, and relative standard deviation of the prediction (RSDP) [29]. The formulas of RMSEC, RMSEP, and RSDP are, respectively:(1)$$\mathrm{RMSEC}=\sqrt{\frac{\sum_{1}^{m}{\left(y_{i,actual}-y_{i,predicted}\right)}^{2}}{m-1}}$$
(2)$$\mathrm{RMSEP}=\sqrt{\frac{\sum_{1}^{n}{\left(y_{i,actual}-y_{i,predicted}\right)}^{2}}{n-1}}$$
(3)$$\mathrm{RSDP}\left(\%\right)=\frac{RMSEP}{\bar{y}}=\frac{\sqrt{\frac{\sum_{1}^{n}{\left(y_{i,actual}-y_{i,predicted}\right)}^{2}}{n-1}}}{\bar{y}}\times 100\%$$
where *m* and *n* are the numbers of calibration and prediction sets, respectively; $y_{i,actual}$ and $y_{i,predicted}$ are the measured and predicted value, respectively; and $\bar{y}$ is the averaged value of the prediction set. A lower value of RSDP (or a higher value of *R*^2^_p_ and a lower value of RMSEP) signifies a better modeling effect. In addition, a *P* value of the pairwise *t*-test higher than 0.05 (paired *t*-test at a 95% confidence level) signifies that there is no significant difference between the measurement of LIBS and ICP-MS [42].

## 4. Conclusions

In this work, we investigated the feasibility of applying LIBS technology for the quantitative analysis of major nutritional and heavy metals in agricultural biochar. Approximately thirty emission lines of the major metal atoms and ions were confirmed based on the NIST database. Univariate analysis based on the sensitive emission lines could not meet the requirement of simultaneous and accurate detection of the major metals in agricultural biochar. In comparison with PLSR and traditional BP-ANN, the nonlinear BP-Adaboost had a better performance for all metal elements—Pb, Cr, Cu, Zn, K, Na, Ca, and Mg—resulting in *R*^2^_p_ values of 0.8497, 0.9463, 0.9584, 0.9798, 0.9838, 0.9388, 0.5280, and 0.9562, and RSDP values of 13.33%, 10.18%, 5.00%, 17.40%, 3.42%, 12.54%, 8.23%, and 8.42%, respectively. Furthermore, the pairwise *t*-test of the prediction set showed that there was no significant difference between the measurement of LIBS and ICP-MS. The LIBS technology has potential as a fast and minimally destructive method for accurately and simultaneously predicting the multiple functional attributes of agricultural biochar samples, providing technical reference for the development of online LIBS equipment.

## Figures and Tables

**Figure 1 molecules-24-03753-f001:**
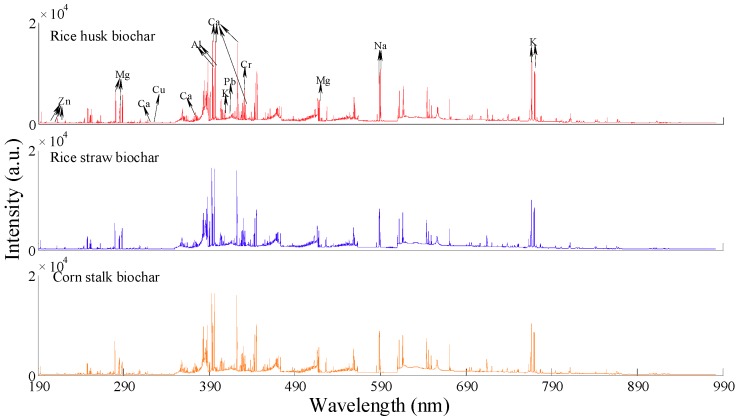
Averaged LIBS spectra of agricultural biochar.

**Figure 2 molecules-24-03753-f002:**
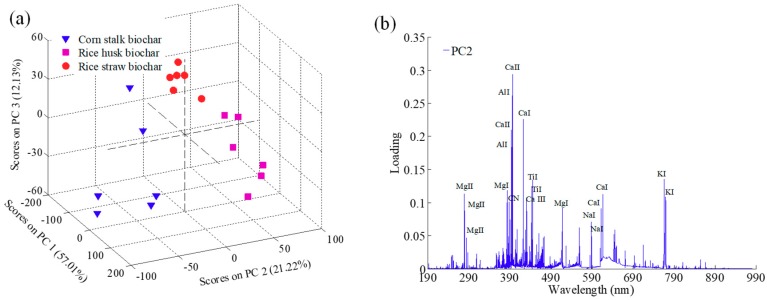
Score plot (**a**) and loading plot (**b**) of the first three principal components (PCs).

**Figure 3 molecules-24-03753-f003:**
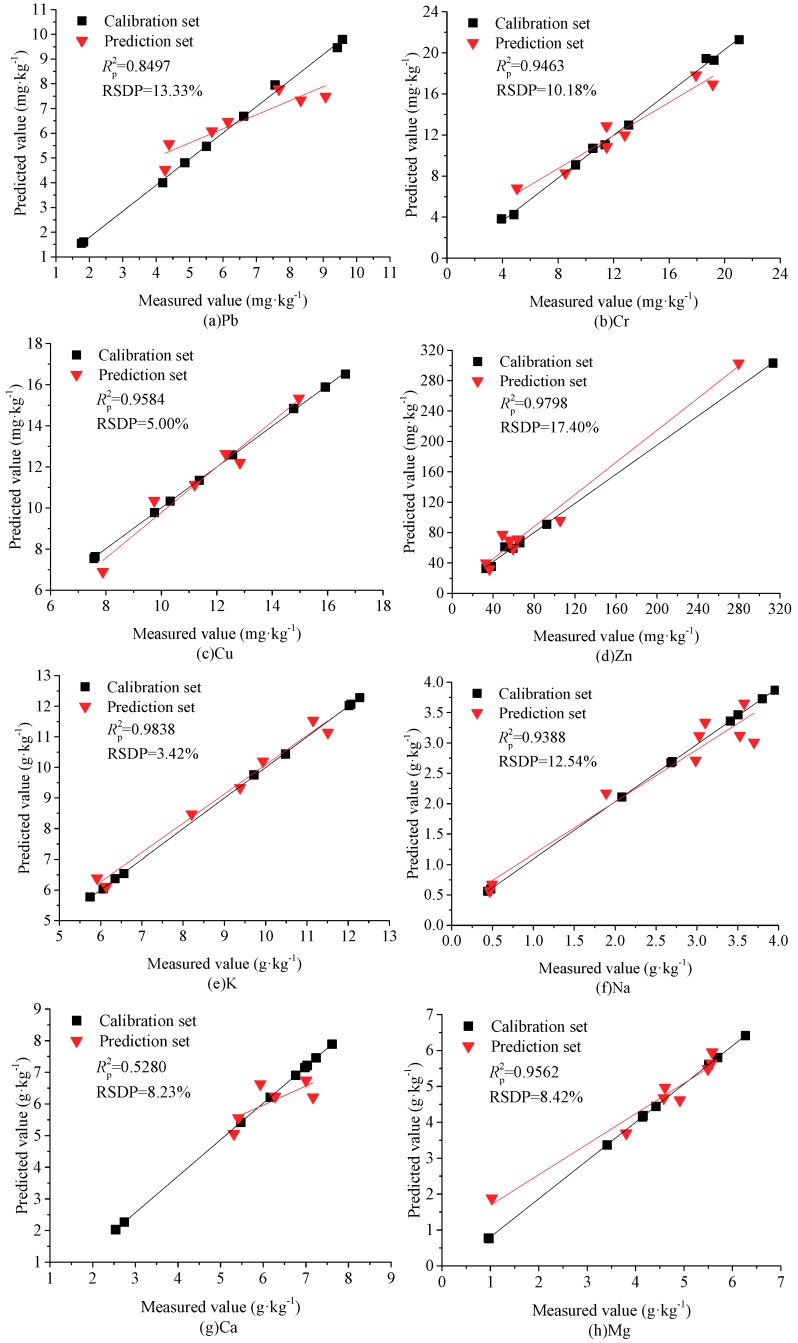
Prediction results of major metals in agricultural biochar using BP-Adaboost. (**a**) Pb, (**b**) Cr, (**c**) Cu, (**d**) Zn, (**e**) K, (**f**) Na, (**g**) Ca, (**h**) Mg.

**Figure 4 molecules-24-03753-f004:**
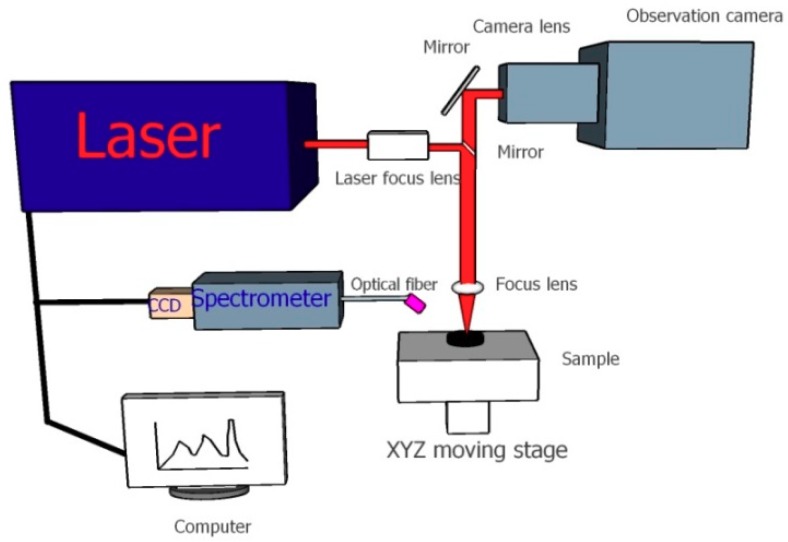
Schematic diagram of the laser-induced breakdown spectroscopy (LIBS) system.

**Figure 5 molecules-24-03753-f005:**
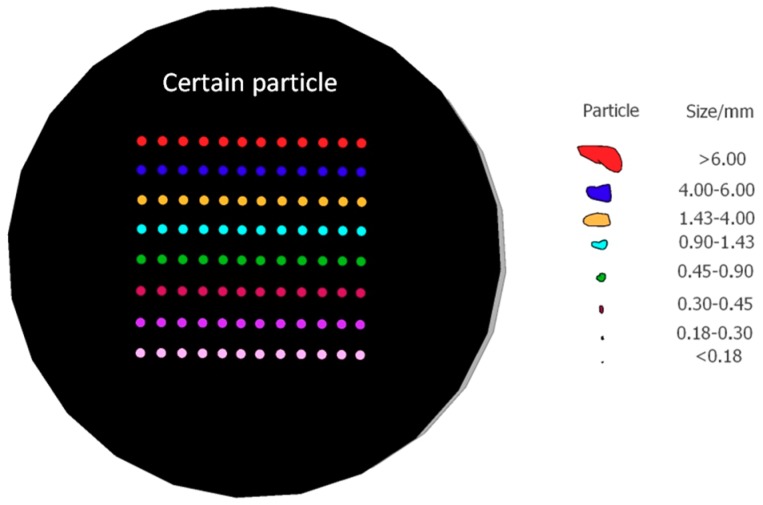
Spectral acquisition of standard samples.

**Table 1 molecules-24-03753-t001:** Statistics results of prepared standard samples.

Element	Calibration Set			Prediction Set		
	Sample	Range	Mean ± SD	Sample	Range	Mean ± SD
Pb (mg·kg^−1^)	S1–S9	1.76–9.58	5.70 ± 2.89	S10, S11, S13, S14, S16–S18	4.27–9.07	6.51 ± 1.90
Cr (mg·kg^−1^)	S1–S9	3.93–21.04	12.44 ± 6.17	S10–S14, S17, S18	5.05–19.15	12.36 ± 4.95
Cu (mg·kg^−1^)	S1–S9	7.56–16.64	11.83 ± 3.39	S10–S14, S17	7.89–14.96	11.49 ± 2.48
Zn (mg·kg^−1^)	S1–S9	32.89–313.29	83.66 ± 88.21	S11–S18	33.12–279.73	85.42 ± 81.60
K (g·kg^−1^)	S1–S9	5.74–12.28	9.03 ± 2.83	S11–S15, S17, S18	5.91–11.51	8.89 ± 2.24
Na (g·kg^−1^)	S1–S9	0.44–3.96	2.56 ± 1.33	S10–S18	0.47–3.70	2.53 ± 1.28
Ca (g·kg^−1^)	S1–S9	2.54–7.61	5.84 ± 1.92	S10, S11, S14, S16–S18	5.31–7.17	6.19 ± 0.78
Mg (g·kg^−1^)	S1–S9	0.96–6.27	3.95 ± 1.91	S10, S11, S13–S18	1.03–5.58	4.45 ± 1.51

**Table 2 molecules-24-03753-t002:** Emission lines of agricultural biochar based on the database of NIST.

Element	Spectral Line (nm)	Peak Broadening Wavebands (nm)
Pb	406.21	405.13–406.64
Cr	427.11, 427.48, 428.27, 428.87	426.50–429.50
Cu	324.75, 327.40	324.50–325.00, 327.25–327.50
Zn	202.55, 206.20, 213.85	202.30-202.80, 206.00–206.40, 213.70–214.10
K	404.41, 404.72, 766.49, 769.90	403–406, 764.5–771.5
Na	588.99, 589.59	588.1–590.2
Ca	315.88, 317.91, 370.56, 373.68, 431.84	315.03–318.92, 370.05–371.07, 373.01–374.02, 431.00–432.06
Mg	279.54, 279.80, 280.27, 285.21, 516.73, 517.27, 518.36	279.30–285.40, 514.00–518.70

**Table 3 molecules-24-03753-t003:** Model results of univariate calibration curve method.

Element	Emission Lines (nm)	Calibration Set		Prediction Set		
		*R* ^2^ _cal_	RMSEC	*R* ^2^ _p_	RMSEP	RSDP (%)
Pb (mg·kg^−1^)	Peak intensity	0.7633	1.6902	0.0492	1.7565	26.98
	Peak area	0.7581	1.3390	0.3133	1.6216	24.91
Cr (mg·kg^−1^)	Peak intensity	0.7549	3.5013	0.5384	3.5371	28.63
	Peak area	0.8870	2.0574	0.7429	2.8936	23.42
Cu (mg·kg^−1^)	Peak intensity	0.8717	1.4727	0.5749	1.5985	13.91
	Peak area	0.9593	0.6451	0.8392	1.1315	9.85
Zn (mg·kg^−1^)	Peak intensity	0.8373	33.5444	0.8733	30.2416	35.40
	Peak area	0.9550	17.6452	0.9502	23.8503	27.92
Ca (g·kg^−1^)	Peak intensity	0.6015	1.1433	0.0779	1.2821	20.72
	Peak area	0.8605	0.6752	0.2947	1.0158	16.42
Mg (g·kg^−1^)	Peak intensity	0.7685	0.9270	0.5158	1.0075	22.65
	Peak area	0.8270	0.7634	0.6214	0.9119	20.50

**Table 4 molecules-24-03753-t004:** Model results of multivariate partial least squares regression (PLSR), backpropagation artificial neural network (BP-ANN), and BP-Adaboost.

Element	Preprocessing	Model	Calibration Set			Prediction Set			
			LVs/PCs	*R* ^2^ _cal_	RMSEC	*R* ^2^ _p_	RMSEP	RSDP (%)	*p* Value
Pb (mg·kg^−1^)	BC + AS	PLSR	2	0.9588	0.5522	0.5260	1.2275	18.86	0.8388
	BC + AS	BP-ANN	4	1.0000	0.1560	0.7739	1.0061	15.46	0.6508
	BC + AS	BP-Adaboost	4	0.9982	0.2152	0.8497	0.8677	13.33	0.9054
Cr (mg·kg^−1^)	BC + Norm + AS	PLSR	2	0.9724	0.9667	0.8494	2.1509	17.41	0.2149
	BC + Norm + AS	BP-ANN	3	0.9988	0.3869	0.9203	1.4368	11.63	0.9836
	BC + Norm + AS	BP-Adaboost	3	0.9980	0.5263	0.9463	1.2574	10.18	0.8228
Cu (mg·kg^−1^)	None	PLSR	6	0.9981	0.1378	0.9421	0.6461	5.62	0.8317
	None	BP-ANN	14	1.0000	0.0099	0.8947	1.6909	14.71	0.2296
	None	BP-Adaboost	14	0.9998	0.0539	0.9584	0.5751	5.00	0.8350
Zn (mg·kg^−1^)	BC	PLSR	5	0.9989	2.7311	0.9866	15.6310	18.30	0.3171
	BC	BP-ANN	16	1.0000	0.0086	0.9623	15.1244	17.71	0.6623
	BC	BP-Adaboost	16	0.9980	5.0042	0.9798	14.8650	17.40	0.1212
K (g·kg^−1^)	None	PLSR	4	0.9344	0.6851	0.8723	1.1060	12.44	0.0954
	None	BP-ANN	10	1.0000	0.0177	0.9735	0.6854	7.71	0.0431
	None	BP-Adaboost	10	0.9999	0.0262	0.9838	0.3038	3.42	0.2640
Na (g·kg^−1^)	BC + Norm + AS	PLSR	4	0.9980	0.0568	0.8919	0.4256	16.82	0.6588
	BC + Norm + AS	BP-ANN	7	1.0000	0.0752	0.8777	0.4521	17.86	0.3664
	BC + Norm + AS	BP-Adaboost	7	1.0000	0.0728	0.9388	0.3174	12.54	0.6907
Ca (g·kg^−1^)	BC + Norm + AS	PLSR	2	0.9251	0.4949	0.5058	0.7066	11.42	0.1985
	BC + Norm + AS	BP-ANN	11	0.9993	0.1141	0.5657	0.6093	9.85	0.9502
	BC + Norm + AS	BP-Adaboost	11	1.000	0.1038	0.5280	0.5090	8.23	0.6367
Mg (g·kg^−1^)	Norm + AS	PLSR	4	0.9829	0.2359	0.9400	0.4081	9.17	0.7353
	Norm + AS	BP-ANN	6	1.0000	0.1077	0.9113	0.4455	10.02	0.4571
	Norm + AS	BP-Adaboost	6	1.0000	0.1093	0.9562	0.3744	8.42	0.2213

**Table 5 molecules-24-03753-t005:** Related literature of detection of main metals in soils using LIBS.

Particle	Element	RSDP (%)	Remarks	Ref.
Soil	Pb, Cr, Cu	18.092, 11.460, 11.956	Lasso ^1^, PCR ^2^	Wang et al., 2018 [31]
Soil	Pb	13.529	PLSR	Yu et al., 2016 [32]
Soil	Cr, Cu	17.673, 18.304	MIPW-PLS	Duan et al., 2018 [33]
Soil	Cr, Cu, Ca, Mg	23.019, 21.682, 33.063, 25.427	MIPW-PLS ^3^	Fu et al., 2017 [34]
Soil	Cu	10.496	ANN ^4^	Ferreira et al., 2008 [35]
Soil	K	5.49	CNN ^5^	Lu et al., 2018 [36]
Soil	K	9.26	Internal standard reference	Dong et al., 2013 [37]
Soil	Zn	18.73	Kriging interpolation method	Kim et al., 2014 [38]

^1^ Lasso: least absolute shrinkage and selection operator; ^2^ PCR: principal component regression; ^3^ MIPW-PLS: modified iterative predictor weighting–partial least squares; ^4^ ANN: artificial neural network; ^5^ CNN: convolutional neural network.

**Table 6 molecules-24-03753-t006:** Elemental content of major metals in standard samples.

Number	Pb (mg·kg^−1^)	Cr (mg·kg^−1^)	Cu (mg·kg^−1^)	Zn (mg·kg^−1^)	K (g·kg^−1^)	Na (g·kg^−1^)	Ca (g·kg^−1^)	Mg (g·kg^−1^)
S1	1.76 ± 0.79	4.83 ± 1.72	7.56 ± 0.45	92.38 ± 63.17	10.48 ± 0.78	0.48 ± 0.03	2.74 ± 0.59	0.97 ± 0.13
S2	1.83 ± 0.55	3.93 ± 2.18	7.60 ± 0.86	66.00 ± 41.46	9.72 ± 0.65	0.44 ± 0.06	2.54 ± 0.39	0.96 ± 0.11
S3	4.20 ± 0.77	9.27 ± 4.18	9.75 ± 0.66	66.15 ± 35.19	6.06 ± 0.97	2.08 ± 0.59	5.47 ± 0.91	3.41 ± 0.48
S4	4.86 ± 1.01	11.40 ± 2.79	10.32 ± 1.04	32.89 ± 4.87	6.35 ± 1.09	2.70 ± 0.82	7.03 ± 0.78	4.16 ± 0.34
S5	5.51 ± 0.80	10.52 ± 2.07	11.37 ± 1.60	38.16 ± 5.93	6.57 ± 1.53	2.68 ± 0.63	7.24 ± 2.17	4.14 ± 0.61
S6	6.62 ± 0.63	13.09 ± 2.21	12.57 ± 0.92	33.23 ± 1.69	5.74 ± 1.25	3.41 ± 0.91	6.76 ± 0.84	4.42 ± 0.41
S7	7.56 ± 1.15	18.66 ± 3.10	14.77 ± 1.69	313.29 ± 170.83	12.07 ± 0.51	3.51 ± 0.54	6.98 ± 0.80	5.69 ± 0.61
S8	9.43 ± 1.49	21.04 ± 4.78	16.64 ± 2.03	51.30 ± 9.70	12.28 ± 0.84	3.96 ± 0.34	7.61 ± 1.67	6.27 ± 1.12
S9	9.58 ± 1.22	19.23 ± 3.04	15.92 ± 1.19	59.58 ± 18.81	12.02 ± 0.87	3.80 ± 0.78	6.16 ± 0.74	5.51 ± 0.66
S10	1.40 ± 0.37	5.05 ± 1.93	7.89 ± 0.56	105.42 ± 28.50	9.94 ± 0.78	0.49 ± 0.04	2.50 ± 0.61	0.87 ± 0.08
S11	1.69 ± 0.64	3.32 ± 2.66	7.56 ± 0.59	63.94 ± 15.94	9.38 ± 0.79	0.47 ± 0.03	2.42 ± 0.38	1.03 ± 0.08
S12	4.27 ± 0.90	8.53 ± 5.29	9.74 ± 0.75	59.67 ± 41.04	5.91 ± 0.88	1.89 ± 0.76	5.31 ± 1.00	3.81 ± 0.50
S13	4.39 ± 1.10	11.50 ± 3.04	11.20 ± 1.20	33.12 ± 4.62	6.14 ± 1.23	2.99 ± 1.01	7.17 ± 0.59	4.58 ± 0.43
S14	5.67 ± 0.94	12.81 ± 2.71	12.84 ± 1.98	36.40 ± 7.13	8.21 ± 2.06	3.53 ± 0.85	7.91 ± 2.36	4.92 ± 0.83
S15	6.15 ± 0.64	11.51 ± 2.66	12.32 ± 0.70	32.66 ± 1.63	5.55 ± 1.56	3.10 ± 1.17	7.01 ± 0.34	4.61 ± 0.31
S16	7.68 ± 1.15	19.15 ± 3.65	17.41 ± 2.09	279.73 ± 75.42	11.15 ± 0.46	3.03 ± 0.65	5.93 ± 0.55	5.58 ± 0.56
S17	8.33 ± 1.43	17.94 ± 3.55	14.96 ± 1.31	55.98 ± 7.48	11.51 ± 0.79	3.58 ± 1.01	5.42 ± 0.99	5.56 ± 0.37
S18	9.07 ± 1.26	21.25 ± 2.80	17.35 ± 1.41	49.08 ± 5.44	12.65 ± 0.99	3.70 ± 0.20	6.28 ± 0.77	5.49 ± 0.82

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
