# Peer review of "Quantitative Analysis of Major Metals in Agricultural Biochar Using Laser-Induced Breakdown Spectroscopy with an Adaboost Artificial Neural Network Algorithm"

_molecules, 2019, doi:10.3390/molecules24203753_

Round 1

Reviewer 1 Report

The paper has very important information and to develop a multiparameter methodology is highly desirable. I think is very well written and explained. Only little observations were done to the manuscript. 

Author Response

Dear Editors and Reviewers:

Thank you for your letter and for the reviewers’ comments concerning our manuscript entitled “Quantitative analysis of major metals in agricultural biochar using laser-induced breakdown spectroscopy with an adaboost artificial neural network algorithm”(molecules-612071). Those comments are all valuable and very helpful for revising and improving our paper, as well as the important guiding significance to our researches. We have studied comments carefully and have made correction which we hope meet with approval. Revised portion are marked in red in the paper. The main corrections in the paper and the response to the reviewer’s comments are as following:

Reviewer 1 Comments

The paper has very important information and to develop a multiparameter methodology is highly desirable. I think is very well written and explained. Only little observations were done to the manuscript. I strongly suggest its publication.

Abstract section

Point 1: Line 10 to 11: I consider the sentence is confusing, I barely understand the meaning, please rewrite.

Response 1: We have rewritten this sentence as “To promote the green development of agriculture by “returning biochar to farmland”, it is great of significance to simultaneously detect the heavy and nutritional metals in agricultural biochar”, in page 1, lines 10-12.

Point 2: Line 16: Why the performance hardly satisfy the requirements of practical application?

Response 2: For the two types of univariate models, the correlation between the measured values and the predicted values was poor, and the relative standard deviation of the prediction (RSDP) values for Pb, Cr, Zn and Mg were all very large (20%-40%). Therefore, the two types of univariate models could not achieve the goal of simultaneous quantitative analysis of multi-indicators in agricultural biochar, which thus hardly satisfy the requirements of practical application.

Point 3:. Line 20: The acronym RSDP was not defined previously.

Response 3: Special thanks to you for your good suggestions, the acronym RSDP was supplemented and defined in Page 1, lines 18.

Experimental design and methods section

Point 4: Line 98: What do you mean by six rice husk, rice straw and corn stalk? You bought six batches of these products or the rice husk and straw and the corn stalk are already in the batches of biochar

product?

Response 4: It means that we bought six batches of these products.

Point 5: Line 113: How do you prepare the digestion solution?

Response 5: Approximately 0.50 g of each sample was mixed with 7.00 mL of a concentrated nitric acid solution and was then placed in a closed microwave digestion system (Milestone Touch, Italy) for digestion. After the acid was driven out on a hot plate at 160 °C, the solution was transferred into a 100.00mL volumetric flask to provide a constant volume. The digestion solution was then prepared.

Point 6: Line 115: The prepared samples for LIBS are only the biochar samples sieved and crushed or is there a previous treatment?

Response 6: The prepared samples for LIBS are only the biochar samples sieved and crushed.

Point 7: Line 124: Table 1 is a very good explained data, but I suggest the use of 2 numbers after the period in the average concentrations. Also I think is suitable to add the standard deviation only to realize the data dispersion.

Response 7: 2 numbers after the period in the average concentrations were used and the standard deviation was added as shown in Table 1.

Point 8: Line 126 to 131: This paragraph is repeated in lines 97-102.

Response 8: We are very sorry for our negligence of this repeted paragraph in lines 126-131, which was deleted in the revised manuscript.

Point 9: Line 156 to 157: The sentence “in addition….” is not quite clear, please rewrite.

Response 9: We have rewritten this sentence as “In addition, the P value of the pairwise T test higher than 0.05 (paired t-test at a 95% confidence level) signifiers that there is no significant difference between the measurement of LIBS and ICP-MS”, in page 6, lines 153-155.

Results section

Point 10: Line 162 to 163: why some samples were selected as the calibration set and other as prediction set? For these analyses were not used the samples divided as particle size?

Response 10: The calibration set was used to develop the calibration model, while the prediction set was used to validate the performance of the calibration model. Moreover, the averaged spectrum and elemental content of the samples divided as particle size were taken as the standard references, which were then selected as the calibration set and prediction set.

Point 11: Line 238 and 239: I suggest to add the year in parenthesis next to the references.

Response 11: The year in parenthesis next to the references was added in page 9, line 237-238, 244.

Point 12: Line 266: I suggest to add the year in parenthesis next to the reference.

Response 12: The year in parenthesis next to the references was added in page 9, line 266.

We tried our best to improve the manuscript and made some changes marked in red font in the revised manuscript. We appreciate for Editors/Reviewers’ warm work earnestly, and hope that the correction will meet with approval. Once again, thank you very much for your comments and suggestions.

Best regards.

Yours sincerely,

Hongwei Duan

Emai: dhwsg123@cau.edu.cn

Guangqun Huang

Email: huanggq@cau.edu.cn

Reviewer 2 Report

The paper submitted by Duan et al. is potentially interesting, although in many parts the chemometric analysis of the LIBS spectra seems to have received more attention than the correct acquisition of the same spectra. The quality of the analytical results would have benefitted of a more careful planning of the spectral acquisition strategy. The description of the LIBS technique is questionable (the meaning of the sentence: "The scanner sequentially obtained the spectral information of each point in the range of 187.78-982.29 nm" is obscure), matrix effect is mistaken with the effect of self-absorption (page 6), the first paragraph of section 2.2 is repeated at the beginning of paragraph 2.3. About the main result of the manuscript, based on the application of the adaboost algorithm, one should firstly ask whether the mere use of an algorithm (in Matlab, I assume) would justify writing a research paper. At least, a comparison of the results with the ones of a conventional ANN should be presented, to justify the use of the proposed method.

Author Response

Dear Editors and Reviewers:

Thank you for your letter and for the reviewers’ comments concerning our manuscript entitled “Quantitative analysis of major metals in agricultural biochar using laser-induced breakdown spectroscopy with an adaboost artificial neural network algorithm”(molecules-612071). Those comments are all valuable and very helpful for revising and improving our paper, as well as the important guiding significance to our researches. We have studied comments carefully and have made correction which we hope meet with approval. Revised portion are marked in red in the paper. The main corrections in the paper and the response to the reviewer’s comments are as following:

Reviewer 2 Comments

The paper submitted by Duan et al. is potentially interesting, although in many parts the chemometric analysis of the LIBS spectra seems to have received more attention than the correct acquisition of the same spectra. The quality of the analytical results would have benefitted of a more careful planning of the spectral acquisition strategy.

Point 1: The description of the LIBS technique is questionable (the meaning of the sentence: "The scanner sequentially obtained the spectral information of each point in the range of 187.78-982.29 nm" is obscure).

Response 1: Special thanks to you for your good suggestions, we have rewritten this sentence as "The spectra of multiple spots on sample surface were acquired, and the averaged spectrum was determined as the final spectrum of each sample", in page 2, lines 90-92.

Point 2: Matrix effect is mistaken with the effect of self-absorption (page 6).

Response 2: The mistake was corrected in page 5 in the revised manuscript.

Point 3: The first paragraph of section 2.2 is repeated at the beginning of paragraph 2.3

Response 3: We are very sorry for our negligence of this repeted paragraph at the beginning of paragraph 2.3, which was deleted in the revised manuscript.

Point 4: About the main result of the manuscript, based on the application of the adaboost algorithm, one should firstly ask whether the mere use of an algorithm (in Matlab, I assume) would justify writing a research paper. At least, a comparison of the results with the ones of a conventional ANN should be presented, to justify the use of the proposed method.

Response 4: Considering the Reviewer’s good suggestion, the BP-ANN models for heavy and nutritional metals in agricultural biochar were developed, and the performances were compared with the PLSR and BP-Adaboost models, as presented in Table 5.

We tried our best to improve the manuscript and made some changes marked in red font in the revised manuscript. We appreciate for Editors/Reviewers’ warm work earnestly, and hope that the correction will meet with approval. Once again, thank you very much for your comments and suggestions.

Best regards.

Yours sincerely,

Hongwei Duan

Emai: dhwsg123@cau.edu.cn

Guangqun Huang

Email: huanggq@cau.edu.cn

Round 2

Reviewer 2 Report

The authors have replied satisfactorily to all the reviewer's remarks. The manscript should now be considered for publication on Molecules.